# Physician preparedness for resource allocation decisions under pandemic conditions: A cross-sectional survey of Canadian physicians, April 2020

Brian Dewar[1], Joanna E. Anderson[2], Edmund S. H. Kwok[3], Tim Ramsay[1], Dar Dowlatshahi[1,4], Robert Fahed[1,4], Claire Dyason[4], Michel Shamy ◉[1,4]*

1 Department of Neurology, The Ottawa Hospital Research Institute, Ottawa, Ontario, Canada,
2 Department of Psychology, Carleton University, Ottawa, Ontario, Canada, 3 Department of Emergency Medicine, University of Ottawa, Ottawa, Ontario, Canada, 4 Department of Medicine, University of Ottawa, Ottawa, Ontario, Canada

* mshamy@toh.ca

**Data Availability Statement:** As our data include responses that are potentially identifiable due to low numbers in some categories, we have chosen

## Abstract

### Background

Under the pandemic conditions created by the novel coronavirus of 2019 (COVID-19), physicians have faced difficult choices allocating scarce resources, including but not limited to critical care beds and ventilators. Past experiences with severe acute respiratory syndrome (SARS) and current reports suggest that making these decisions carries a heavy emotional toll for physicians around the world. We sought to explore Canadian physicians' preparedness and attitudes regarding resource allocation decisions.

### Methods

From April 3 to April 13, 2020, we conducted an 8-question online survey of physicians practicing in the region of Ottawa, Ontario, Canada, organized around 4 themes: physician preparedness for resource rationing, physician preparedness to offer palliative care, attitudes towards resource allocation policy, and approaches to resource allocation decision-making.

### Results

We collected 219 responses, of which 165 were used for analysis. The majority (78%) of respondents felt "somewhat" or "a little prepared" to make resource allocation decisions, and 13% felt "not at all prepared." A majority of respondents (63%) expected the provision of palliative care to be "very" or "somewhat difficult." Most respondents (83%) either strongly or somewhat agreed that there should be policy to guide resource allocation. Physicians overwhelmingly agreed on certain factors that would be important in resource allocation, including whether patients were likely to survive, and whether they had dementia and other significant comorbidities. Respondents generally did not feel confident that they would have the social support they needed at the time of making resource allocation decisions.

to limit access by making our data available upon request. The data underlying the results presented in the study are available from Zenodo: Shamy, Michel; Dewar, Brian. (2020). Dataset for Physician Preparedness for Resource Allocation Decisions Under Pandemic Conditions: A Cross-Sectional Survey of Canadian Physicians, April 2020 (Version 1) [Data set]. Zenodo. http://doi.org/10.5281/zenodo.4008146.

**Funding:** The project was funded by University of Ottawa Department of Medicine Special Pandemic Agile Research Competition (SPARC) Grant. The funders had no role in study design, data collection and analysis, decision to publish, or preparation of the manuscript.

**Competing interests:** The authors have declared that no competing interests exist.

## Interpretation

This rapidly implemented survey suggests that a sample of Canadian physicians feel under-prepared to make resource allocation decisions, and desire both more emotional support and clear, transparent, evidence-based policy.

## Introduction

Under the pandemic conditions created by COVID-19, physicians around the world have faced difficult choices around the allocation of scarce resources, including but not limited to critical care beds and ventilators [1, 2]. Making these decisions has left Italian physicians "weeping in hospital hallways," [3] and emerging reports suggest that American physicians have been similarly affected [4] by the emotional toll of the "toughest triage" [5]. One such resource specific to this region, predicted a worst-case scenario of more than 13,000 Ontario patients left to die due to insufficient resources [6] and another predicted the possibility of an insufficiency of intensive care unit (ICU) beds [7]. While many ethical frameworks exist for allocating resources in pandemics [8–11], we have few empirically-based insights into physicians' attitudes and beliefs surrounding resource allocation decisions in the era of COVID-19 [12].

In early April 2020, we launched a survey of physicians practicing in the region of Ottawa, Ontario, Canada. Ottawa is the national capital, a cosmopolitan and bilingual city of approximately 1 million. We sought to capture physician preparedness to make resource allocation decisions, their anticipated approach to these decisions, their awareness of existing guidelines, their comfort with the provision of palliative care under pandemic circumstances, and their desire for services to support their decision-making. We hypothesized that our respondents would not feel prepared to make these decisions, would be unaware of any specific guidelines, and would use commonly cited factors such as age and comorbidities to make resource allocation decisions.

We timed our survey to predate the expected surge of COVID-related hospitalizations in our region. At the time the survey was launched, there were approximately a dozen patients with COVID-19 admitted to The Ottawa Hospital, the academic tertiary care centre in our region. We kept the survey open only for only 10 days so as to capture a specific moment in time when resource allocation was a foremost concern. As a reflection of how quickly the field was moving, the province of Ontario issued a guideline on triaging access to critical care beds during the time we were designing the survey [13]. Under hypothetical "surge" conditions that were ultimately never met, patients would be excluded from critical care according to disease-specific mortality thresholds that become increasingly more stringent as resources become more limited.

## Methods

Data for this study were collected via an online survey (S1 Appendix) administered via the Qualtrics XM survey platform (Denver, Colorado) [14] between April 3 and 13, 2020. Approval from the Ottawa Health Sciences Network Research Ethics Board (application 20200208-01H (2118)) was sought and obtained. Prior to fielding, the survey was piloted on a convenience sample to determine its length and resolve areas of ambiguity. Survey respondents did not receive any incentive and participation was voluntary. The survey included eight

questions on four main themes: physician preparedness for resource rationing, physician preparedness for palliative care, attitudes towards resource allocation policy, and approaches to resource allocation decision-making.

Responses were entirely anonymous and no identifying information such as IP addresses or emails was collected. The risk of repeat participation was minimized in two ways: First, using an option within the survey platform that prevents duplicate participants with a browser cookie; and second, by removing responses that were less than 40% complete.

The population of interest included staff physicians in the Ottawa region, with a sample frame defined by membership in a Facebook group for local physicians and/or mailing lists belonging to the following groups: the Departments of Medicine, Anesthesia, Critical Care and Emergency Medicine at The University of Ottawa / Ottawa Hospital; the Divisions of Neurology and Palliative Care; the Regional Ethics program; and the Ottawa Hospital Research Institute. This group was selected to capture a broad cross-section of physicians within a defined geographic area.

Analysis was primarily descriptive, but appropriate inferential statistics were performed where comparisons between groups or responses were indicated. Pairwise between-group comparisons were corrected using Tukey's honest significant difference test. For more information about survey procedures, please see S1 Checklist, the completed Checklist for Reporting Results of Internet E-Surveys (CHERRIES) checklist [15]. Responses regarding the content of guidelines were thematically analyzed by two independent reviewers.

## Results

The initial sample included 219 partial and complete responses. Of these, 54 were less than 40% complete and were removed prior to analysis to minimize data duplication. This left a final sample of 165 for analysis (Table 1). The majority of responses (70.3%) came from departmental mailing lists, 29.1% came from the Facebook group and 0.6% from a link for participants referred to the survey by previous respondents.

### Physician preparedness for resource rationing

Respondents were asked, "Imagine that you have two patients who require a ventilator but only one ventilator is available. How prepared do you feel to determine who will receive the ventilator?" (Fig 1). The majority of respondents endorsed being "somewhat" or "a little prepared" (78%). However, more than one in ten (13%) described themselves as "not at all prepared." When analyzed by specialty groups with at least 10 respondents (Fig 2), critical care/anaesthesia physicians described themselves as being significantly more prepared to make decisions on ventilator allocation than all other specialities (all $ps \leq .05$). Family medicine practitioners described themselves as less prepared than all other specialties with the exception of surgery (vs. surgery: $p = .94$; vs. others: all $ps < .05$). No other statistically significant correlations were found with regards to specialty.

Respondents were asked to express in one word how they would feel when making a decision about allocating a ventilator (Fig 2). Responses were coded into 11 categories, with the most common categories being "anxious" (29%), "sad" (19%), and "awful" (12%). Less common responses included "calm" (9%), "resolved" (9%), and "confident" (6%). A parallel question asked participants to imagine their feelings after having made such a decision. Respondents predominantly mentioned feelings of sadness (24%) and guilt (19%), followed by acceptance (12%).

**Table 1. Demographic characteristics of sample.**

|  | Count | Percentage |
|---|---|---|
| **Age** | | |
| Under 35 | 29 | 19% |
| 35–44 | 48 | 31% |
| 45–54 | 45 | 29% |
| 55–64 | 26 | 17% |
| 65+ | 7 | 5% |
| Unspecified | 10 | |
| **Gender** | | |
| Male | 82 | 53% |
| Female | 71 | 46% |
| Non-binary* | 1 | 1% |
| **Speciality** | | |
| Critical Care / Anesthesia | 21 | 14% |
| Emergency Medicine | 27 | 18% |
| Family Medicine | 20 | 13% |
| Laboratory Medicine* | 0 | 0.0% |
| Medicine | 64 | 43% |
| Obstetrics/Gynecology* | 2 | 1% |
| Pediatrics* | 1 | 1% |
| Psychiatry* | 1 | 1% |
| Surgery | 13 | 9% |
| Unspecified | 16 | |
| **Total** | **165** | **100%** |

*Note.* Groups with under 10 respondents (indicated with asterisks above) were excluded from demographic analyses to protect their anonymity.

## Physician preparedness for palliative care

Respondents were asked how they expected to feel about providing palliative care to a patient who had been denied life-saving treatment because of resource allocation (Fig 3). A substantial majority (63%) expected this situation to be "very" or "somewhat difficult." Most respondents (55%) described being "somewhat" or "very comfortable" with having goals of care conversations, though respondents were substantially less comfortable with having a goals of care conversation with the patient's family via telephone or videolink, with 22% of respondent describing themselves as "not at all comfortable" doing this.

## Physicians and resource allocation policy

A slight majority (53%) of respondents were aware of any existing policy about resource allocation in pandemics; of those, 61% were aware of the provincial triage policy that had been released the week the survey was launched. Most respondents (83%) either strongly or somewhat agreed that there should be policy to determine who should receive critical care resources in the event of scarcity. Virtually all participants (96%) stated that they would follow such a policy if it aligned with their own values. However, in the hypothetical case that a policy did not align with their own values, the percentage of respondents who stated that they would follow the policy in all circumstances decreased from 32% to 9%, though the majority (65%) would still follow the policy in "all" or "most circumstances" (Fig 4).

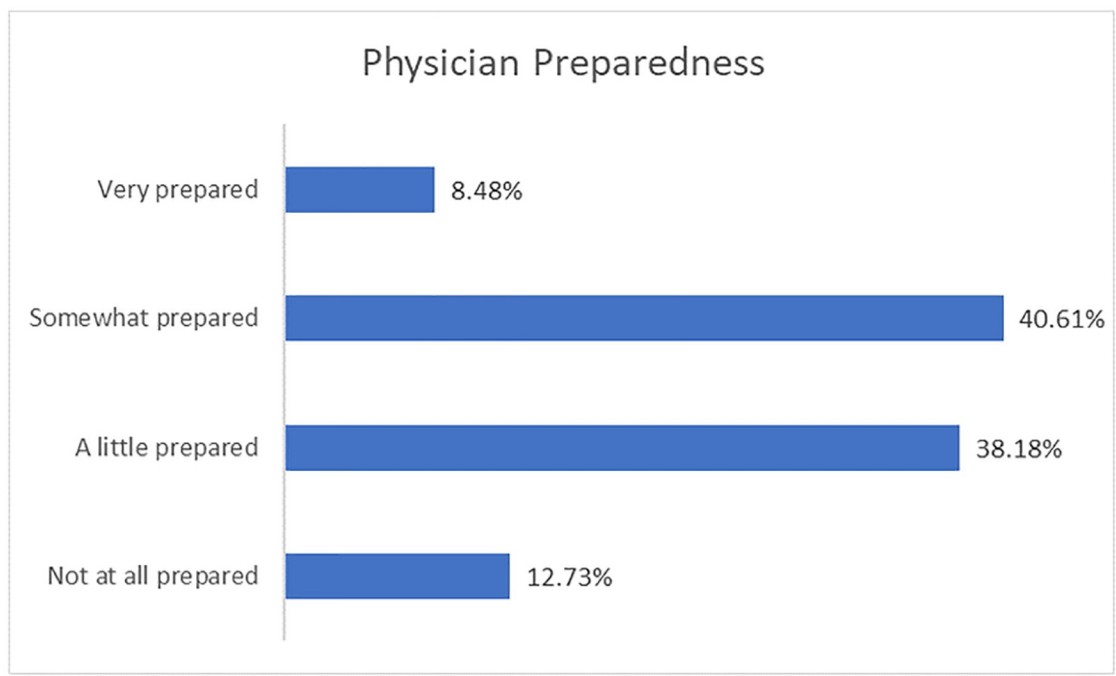

**Fig 1. Physician preparedness.** Self-reported physician preparedness to make resource allocation decisions, measured on a 4-point scale from 1 (not at all prepared) to 4 (very prepared).

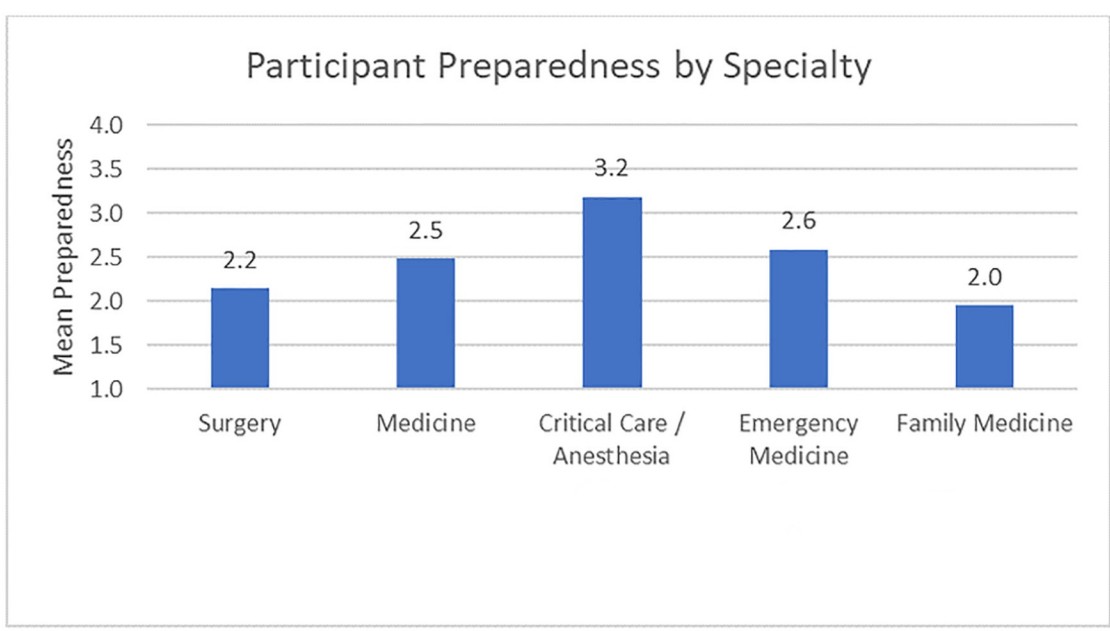

**Fig 2. Physician preparedness by specialty.** Self-reported physician preparedness to make resource allocation decisions by specialty, measured on a 4-point scale from 1 (not at all prepared) to 4 (very prepared). *Note*. Responses were measured on a 4-point Likert-type scale ranging from 1, *not at all prepared*, to 4, *very prepared*.

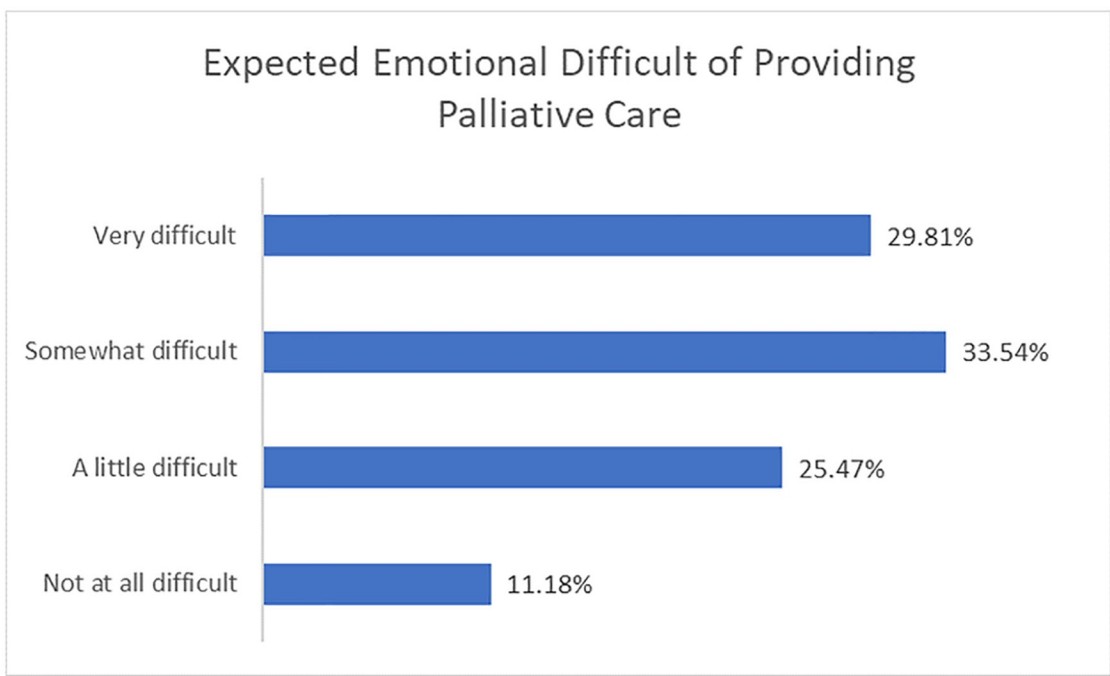

**Fig 3. Expected emotional difficulty of providing palliative care.** Expected emotional difficulty of providing palliative care to patients under pandemic conditions, measured on a 4-point scale from not at all difficult to very difficult.

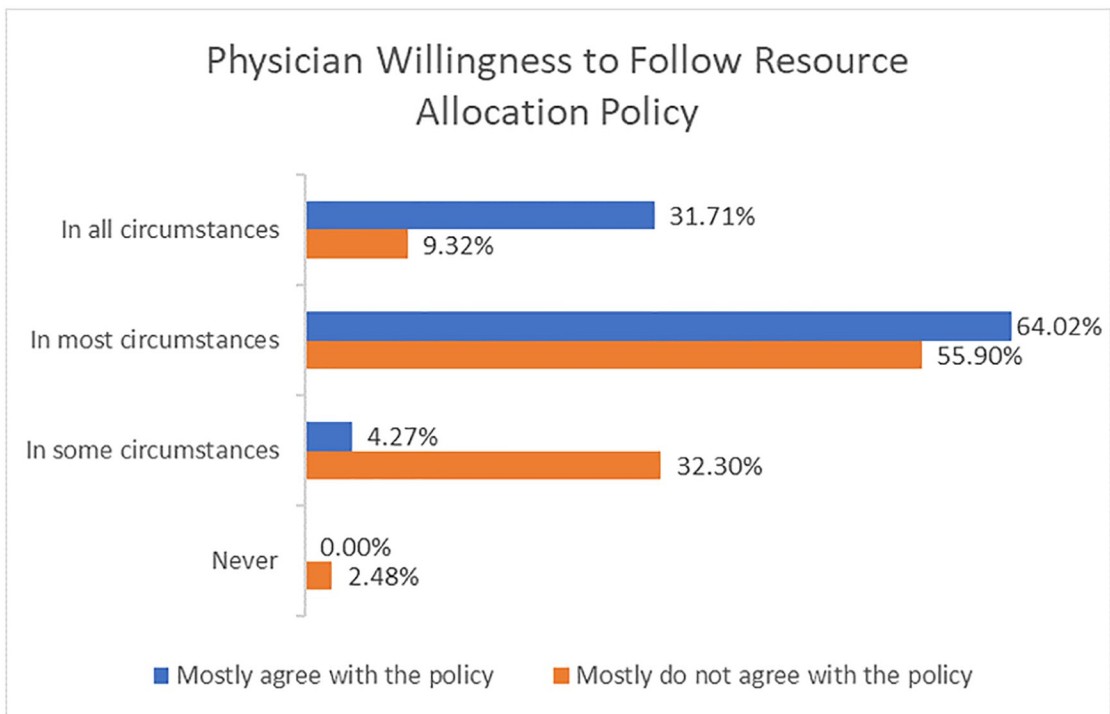

**Fig 4. Physician willingness to follow resource allocation policy.** Physician willingness to follow an institutional policy on resource allocation, measured on a 4-point scale from in all circumstances to never, differentiated by whether the respondent mostly agrees with the policy or mostly does not agree with the policy.

## Policy recommendations

When asked to define what would be most important to include in a policy on resource allocation, explicitness on how to follow it (20%) and transparency in how the policy was designed (18%) emerged as the most common themes. Other frequent responses included some statement on what would be expected of physicians when enacting these policies (9%), the importance of including an evidence-base in the policy (8%), and the importance of addressing issues of legal culpability (3%) (Table 2). Some respondents (20%) wanted to remove decision-making responsibility (and thereby guilt and emotional responsibility) from individual physicians, while other respondents (9%) preferred a more flexible policy that would allow physicians the leeway to make choices in line with their individual values.

## Physician decision-making during resource scarcity

To assess which factors would most strongly influence resource allocation decisions, we asked respondents to make a series of choices about which of two patients should receive a ventilator using a 5-point Likert-type scale with the points *Definitely Patient A*, *Probably Patient A*, *Unsure*, *Probably Patient B*, and *Definitely Patient B*. Each choice differentiated the two patients on just one or two factors (e.g., age, comorbidities), and stated that all else should be considered equal between them.

This choice paradigm was intended to reveal the importance of different patient characteristics for the average respondent. However, we also directly asked respondents to rate the importance of key factors. Thus, these two questions provide us with both revealed and stated importance ratings.

Fig 5 below shows the average likelihood of choosing the patient on the right for each of the choices. Note that 3 was the midpoint ("unsure") of the scale. In all cases, physicians were significantly more likely, on average, to choose the patient characteristics described on the right (all $ts > |3.3|$, $ps < .001$), with one exception: gender ($t[163] = 1.9$, $p = .06$).

However, as the figure makes clear, physicians prioritized survival, cognitive status, comorbidity severity and age when making resource allocation decisions. The mean score for likelihood of survival is close to the maximum value of "definitely Patient B [who has a higher survival chance]", while the age-related item (age 72 versus 40), at 4.0, averages exactly on the scale point of "probably Patient B [the younger patient]."

**Table 2. What is the most important aspect of a policy on resource allocation?**

|  | Count | Percentage |
|---|---|---|
| **Age** |  |  |
| Explicitness | 24 | 20% |
| Transparency | 21 | 18% |
| Statement on Expectations of Physicians | 11 | 9% |
| Ethics Support | 11 | 9% |
| Evidence-Based | 10 | 8% |
| Flexibility | 9 | 8% |
| Preemptoriness | 6 | 5% |
| Inclusiveness | 5 | 4% |
| Consideration of Legal Issues | 4 | 3% |
| Other | 17 | 16% |
| **Total** | **118** | **100%** |

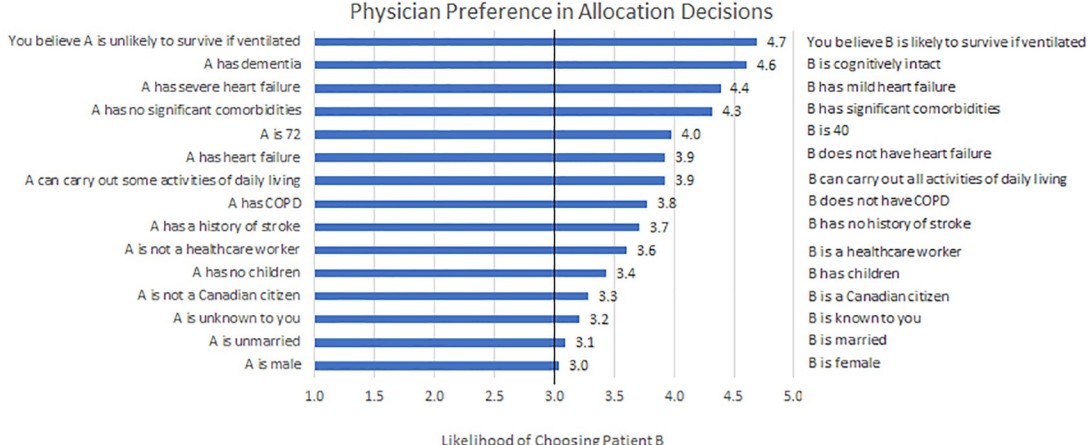

**Fig 5. Physician preference in allocation decisions.** Under hypothetical conditions where there is 1 ventilator for 2 potential patients, physicians were asked to report their preference in allocating this ventilator, all else being equal. Responses were scored on a 5 point scale where 1 was definitely patient A and 5 was definitely patient B. *Notes.* The patient descriptions and means have been reverse-coded in some cases for clearer presentation. The response options were: 1 = *definitely Patient A*, 2 = *probably Patient A*, 3 = *unsure*, 4 = *probably Patient B*, 5 = *definitely Patient B.*

In an attempt to provide a closer behavioural test of how physicians prioritize between some of the key factors such as age and comorbidities, pairings combining these factors (Fig 6) were included. In these cases, respondents tended to prioritize both survival and presence of comorbidities over age.

Finally, participants directly rated the importance of various patient characteristics (Fig 7). Respondents' stated order of importance reflected their approach to the patient comparisons in that likelihood of survival, comorbidities and dementia were the strongest determinants.

## Social support and moral injury

Respondents to this survey were asked if they felt confident that they could access necessary mental health resources at the time of making a resource allocation decision. Among respondents, 38% felt not at all confident, while only 13% felt very confident that they would be able to access these resources. However, a majority (64%) felt either "somewhat confident" or "very

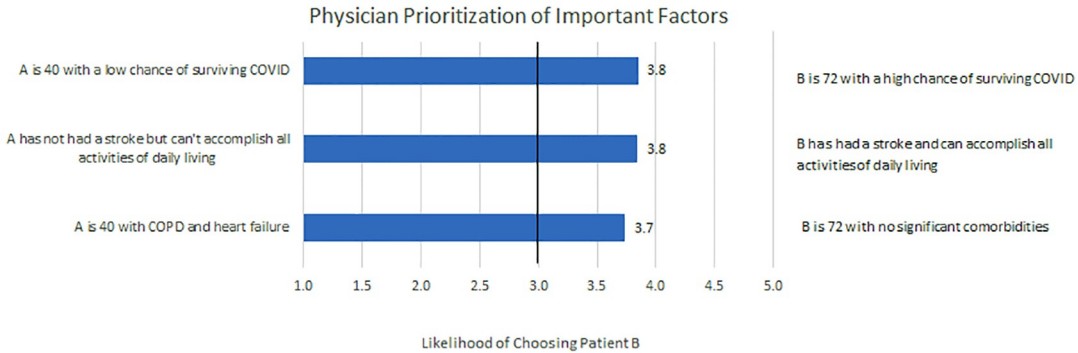

**Fig 6. Physician prioritization of important factors.** Decisions under hypothetical conditions where there is 1 ventilator for 2 potential patients, physicians were asked to report their preference in allocating this ventilator. Responses regarding the impact of age, function and likelihood of survival were cross-compared. *Notes.* The patient descriptions and means have been reverse-coded in some cases for clearer presentation. The response options were: 1 = *definitely Patient A*, 2 = *probably Patient A*, 3 = *unsure*, 4 = *probably Patient B*, 5 = *definitely Patient B.*

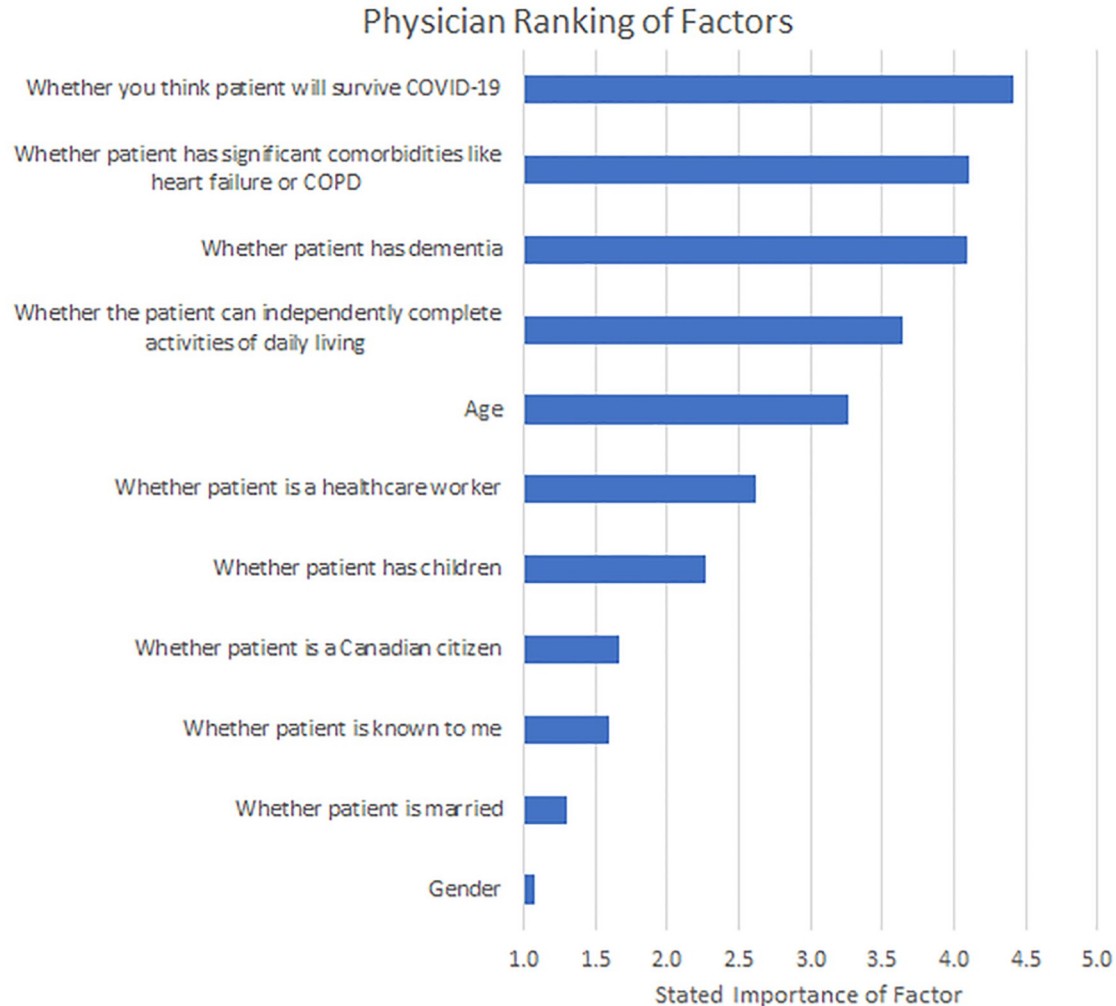

**Fig 7. Physician ranking of factors.** Physicians ranked the factors that they felt would be most impactful on their decision-making regarding allocating ventilators, scored on a 5 point scale where 1 was not at all important and 5 was extremely important. *Note*. The response options were: 1 = *not at all important*, 2 = *slightly important*, 3 = *moderately important*, 4 = *very important*, 5 = *extremely important*.

confident" that they would be able to access mental health resources after making an allocation decision. Physicians reported being most likely to turn to other physicians (44%), followed by family members (28%), professional counsellors (16%), religious advisors (7%) and no one (4%). The most common volunteered responses were turning to hospital ethics support and non-medical friends, both with 4 responses (3%).

## Discussion

This rapidly implemented survey provides insights into physician preparedness to make resource allocation decisions utilizing a series of hypothetical conditions related to COVID-19. the effects of resource insufficiency on physician decision-making, and for healthcare policy-making in pandemics. While this topic has engendered significant debate in the medical litera-ture and popular press in recent weeks, little empirical data has been available to provide context for this debate. Our results provide some insights, albeit from a single Canadian

region, and acquired in a context of anticipation for the COVID surge. We present three major takeaways from this survey:

1. A majority of surveyed physicians reported feeling under-prepared, described themselves as anxious or sad when imagining themselves making tough decisions around allocating resources, and anticipated guilt afterwards. Participants were generally not confident in their ability to access mental health support at the time of making these decisions, though a majority believed they would be able to access support afterwards. Our survey suggests that Canadian physicians are likely to experience similar mental health challenges to colleagues in New York, Italy and China [16] due to COVID-19. Clear guidelines on resource allocation, acknowledgement of the emotional toll of making these decisions, and robust support systems are needed may help alleviate these pressures for physicians worldwide. We suggest that local and national physician organizations should play an important role in supporting physician mental health during this difficult time [17].

2. Respondents demonstrated a strong appetite for transparently-developed, evidence-based, and clear-to-follow guidelines to inform resource allocation decisions. Based on our results, we would encourage institutions to seek to develop documents with a solid, evolving evidence-base, that considers implementation, addresses legal issues, and provides guidance on how to communicate with patients and families.

3. A consistent set of factors emerged as being important to most physicians' decision-making around resource allocation: survival likelihood, cognitive function, comorbidities and daily function. However, significant disagreement existed around other factors including age and citizenship, and our ability to draw conclusions about these factors is limited. In the absence of agreement and standard practices, physicians are at risk of being influenced by unconscious biases in the way resources are rationed [18]. This tendency may be exacerbated when information about the patient or the clinical scenario is limited.

The largest limitation of this work was the fact that participants self-selected into the study rather than being recruited via probability-based sampling. The survey was distributed through several email lists as well as through two Ottawa physician Facebook groups, whose memberships overlap. As such, we are also unable to report an exact response rate, though we estimate that 10–15% of physicians employed by The Ottawa Hospital may have responded within the brief administration period. The goal of the survey was not to provide a perfectly representative picture of staff physicians in Ottawa, but to obtain input that will be useful for the creation of policy and practice. In addition, the sample was regionally limited; however, the Ottawa area is multicultural and has a strong medical academic program. As such, we posit that our results are reasonably generalizable to other regions. Finally, because of the fast-moving nature of the COVID-19 pandemic, policies on resource allocation were being written and released while this survey was recruiting participants, and this may have affected respondent knowledge of extant policies.

Future directions of research raised by this work include systematically reviewing existing frameworks for resource allocation decisions under pandemic conditions, developing methods to raise awareness of existing policies so that physicians are as well-informed as possible, and examining interventions to provide support to physicians at the time of decision-making.

## Supporting information

**S1 Appendix. Preparing for resource rationing under pandemic conditions.**
(DOCX)

**S1 Checklist. Checklist for Reporting Results of Internet E-Surveys (CHERRIES).**
(XLSX)

## Acknowledgments

We would like to thank the participants of this study for their time and expertise during this difficult period. We would also like to thank the following individuals for supporting the distribution of our survey: Dr Phil Wells, Dr Greg Bryson, Dr Scott Millington, Dr Mike Kekewich. Most importantly, we acknowledge the dedication and selflessness of all health care workers facing the challenge of COVID-19 across Canada and around the world.

## Author Contributions

**Conceptualization:** Brian Dewar, Edmund S. H. Kwok, Tim Ramsay, Robert Fahed, Claire Dyason, Michel Shamy.

**Data curation:** Brian Dewar, Joanna E. Anderson, Edmund S. H. Kwok, Claire Dyason, Michel Shamy.

**Formal analysis:** Brian Dewar, Joanna E. Anderson.

**Funding acquisition:** Edmund S. H. Kwok, Tim Ramsay, Robert Fahed, Claire Dyason, Michel Shamy.

**Investigation:** Brian Dewar, Claire Dyason, Michel Shamy.

**Methodology:** Brian Dewar, Joanna E. Anderson, Edmund S. H. Kwok, Tim Ramsay, Dar Dowlatshahi, Robert Fahed, Claire Dyason, Michel Shamy.

**Project administration:** Brian Dewar.

**Supervision:** Michel Shamy.

**Validation:** Dar Dowlatshahi.

**Writing – original draft:** Brian Dewar, Michel Shamy.

**Writing – review & editing:** Brian Dewar, Joanna E. Anderson, Edmund S. H. Kwok, Tim Ramsay, Dar Dowlatshahi, Robert Fahed, Claire Dyason, Michel Shamy.

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
