## [Decision Letter · Decision Letter 0]

15 Jul 2020

PONE-D-20-15128

Physician Preparedness for Resource Allocation Decisions Under Pandemic Conditions: A Cross-Sectional Survey of Ottawa-Area Physicians, April 2020

PLOS ONE

Dear Dr. Shamy,

Thank you for submitting your manuscript to PLOS ONE. After careful consideration, we feel that it has merit but does not fully meet PLOS ONE’s publication criteria as it currently stands. Therefore, we invite you to submit a revised version of the manuscript that addresses the points raised during the review process.

We look forward to receiving your revised manuscript.

Kind regards,

Ritesh G. Menezes, M.B.B.S., M.D., Diplomate N.B.

Academic Editor

PLOS ONE

Journal Requirements:

2.Thank you for including your ethics statement:  "Ottawa Health Sciences Network Research Ethics Board 20200208-01H (2118)"

a) Please amend your current ethics statement to confirm that your named institutional review board or ethics committee specifically approved this study.

Additional Academic Editor Comments:

The methods must be described in sufficient detail for other researchers to reproduce the methodology described. This is a major concern that needs to be addressed while submitting the revised draft of the manuscript. For further details kindly check the following website:

http://journals.plos.org/plosone/s/criteria-for-publication#loc-3

Reviewers' comments:

Reviewer's Responses to Questions

**Comments to the Author**

1. Is the manuscript technically sound, and do the data support the conclusions?

Reviewer #1: Partly

Reviewer #2: Yes

Reviewer #3: Yes

Reviewer #4: Yes

Reviewer #5: Yes

2. Has the statistical analysis been performed appropriately and rigorously? 

Reviewer #1: Yes

Reviewer #2: Yes

Reviewer #3: Yes

Reviewer #4: Yes

Reviewer #5: Yes

3. Have the authors made all data underlying the findings in their manuscript fully available?

Reviewer #1: Yes

Reviewer #2: Yes

Reviewer #3: Yes

Reviewer #4: No

Reviewer #5: Yes

4. Is the manuscript presented in an intelligible fashion and written in standard English?

Reviewer #1: Yes

Reviewer #2: Yes

Reviewer #3: Yes

Reviewer #4: Yes

Reviewer #5: Yes

5. Review Comments to the Author

Reviewer #1: This is a valuable study of the preparedness of physicians with regard to the COVID-19 pandemic. The study employs a mixture of qualitative and quantitative methods to provide a holistic assessment of physicians' subjective perceptions of preparedness at the time immediately preceding the onset of a surge in COVID-19 cases in Canada.

Nonetheless, there are a number of methodological difficulties that should be clarified prior to the publication of this study.

First, the authors have asked research participants the following survey question:

"Imagine that you have two patients who require a ventilator but

only one ventilator is available. How prepared do you feel to determine who will receive the

ventilator"

and further

"how they expected to feel about providing palliative care to a patient

who had been denied life-saving treatment because of resource allocation"

And have inferred from these answers that physicians feel unprepared to respond to the pandemic with their present resources. The difficulty here is that the insufficiency of present resources is one of the conclusions drawn by the researchers, but the lack of sufficient resources is integrated to the premises of the questions (i.e. the physicians are asked to gauge their response to a lack of resources).

I would recommend that the authors reformulate their conclusions such as to clearly demonstrate that they are evaluating the effects of resource insufficiency on physicians in Canada's healthcare sector, rather than evaluating whether physicians consider resource insufficiency to be present prior to the onset of the pandemic.

Further, the researchers have relied on insufficient scholarship to demonstrate the veracity of their premise that resources are insufficiently available to provide care during COVID-19. The majority of the resources cited to this effect are newspaper articles and ethics or decision-science articles that assess how a lack of resources should be addressed. If available, more scholarship demonstrating the lack of healthcare sector resources in the face of Canada's COVID-19 pandemic should be included.

Second, the methodology section (p. 3) is very sparse, and while the methodology selected is clearly described in the ensuing body paragraphs, a more expansive description of the selection criteria, study design, and the perceived strengths and weaknesses of the methods employed would strengthen the article and enable the reader to critically assess how well the study results can be generalized.

The data collected overall provides a nuanced and unique snapshot into the COVID-19 pandemic response and I would heartily recommend publication.

Further, I consider the following conclusion to be overstated or impossible to draw from the data collected:

"Respondents demonstrated a strong appetite for transparently-developed, evidence-based,

and clear-to-follow guidelines to inform resource allocation decisions."

I assume that the above conclusions were drawn from the responses to the questions on pages 5, 6, and 7 of the survey questionnaire. I consider it to be appropriate to draw conclusions therefrom as to the desire of physicians to rely on policy guidelines as opposed to personal, value-driven approaches in making decisions regarding the allocation of limited resources. However, the questions asked in the survey, in my view, do not permit the drawing of conclusions regarding the content that respondents hope to see reflected in such resource-allocation policies.

I note that there is a significant over-representation of laboratory medicine specialists among respondents. It is not required, but add to the depth of insight provided by the study, for respondents to note if any correlations appear among their results regarding the specializations studied. More generally, considerable demographic information appears to have been collected about the participants, but no conclusions are drawn with regard thereto. Resources and space permitting, the authors should consider adding information with regard to the correlation emergent from this data.

Last, the results provided appear to suggest that research participants would slightly favor the allocation of resources to Canadian citizens over non-Canadian citizens. This could have serious implications for the potential for Canada's pandemic response and potential systemic discrimination or bias - the researchers should consider highlighting this conclusion and its implications for pandemic response in their findings.

These caveats aside, the study presents a crucial portrait of a critical moment in the evolution of the COVID-19 pandemic, and both the results collected and the insights drawn therefrom will be of great value for the future of this pandemic response effort and future pandemic response efforts.

Reviewer #2: The authors have performed a cross-sectional survey among Canadian physicians in April 2020 to assess preparedness for necessary decisions to be made regarding resource allocation during the COVID-19 pandemic and beyond. Resource allocation decisions have taken a large toll on physicians around the world, often due to lack of preparedness training for making these decisions during crises and outbreaks. The authors are to be applauded for their important work. I have some comments to improve the manuscript:

Major comments:

1. The authors note that “At the time the survey was launched, there were approximately a dozen patients with COVID-19 admitted to The Ottawa Hospital,” which preceded the anticipated surge. Compared to overwhelmed hospitals in, for example, New York City, Lombardy, Madrid, etc. the survey was conducted at a time where the hospital (which is the largest tertiary center in the region) was far from being overwhelmed, as was intended. Can the authors comment on whether surveys will be conducted later in the outbreak or after the pandemic to assess changes in average perceptions, whether as a result of lived experiences or introduced policy changes, support mechanisms, and/or crash courses supporting these decisions?

2. Interestingly, surgical respondents were found to be among the most ill-prepared in terms of making difficult resource allocation decisions. The authors may want to discuss this point, as 1) many countries include brief critical care training in surgical training and 2) surgical teams are commonly deployed as one of the first services to support COVID-19 care (e.g., because of reduced elective services, to place cathethers or perform tracheostomies, etc.), which may be currently or in the future perceived in Canada as well in case of major surges/outbreaks.

Minor comments:

1. Abstract background and body introduction: write abbreviations (here: SARS and COVID-19) in full when using them for the first time.

2. Methods: “REB”, “HSD”, and “CHERRIES” may be unclear to the reader, write abbreviations in full before using them.

3. Results, line 2: “to prevent data duplication” should be “to minimize data duplication”; including surveys that are 50% complete may still result in duplication.

Reviewer #3: The paper by Dewar and colleagues "Physician Preparedness for Resource Allocation Decisions Under Pandemic

Conditions: A Cross-Sectional Survey of Ottawa-Area Physicians, April 2020" is very well prepared, well reported and well written.

I dont any major points to add. The paper is very important and should be released on timely manner.

The charts can be visually improved using professional software!

Reviewer #4: The manuscript addresses some challenging issues that physicians encounter while managing COVID-19 patients with limited resources. The questionnaire seems adequate. The language overall was adequate. I do recommend the study included only those who completed 66 to 75% of questionnaires.

Reviewer #5: This is an interesting evaluation of results of a survey on preparedness and attitudes regarding resource allocation decisions among Canadian physicians. Although the sample is relatively small, this study provides insight on often overlooked aspects of resource allocation decision-making. The timing of the survey is particularly interesting, as it was administered before the surge of COVID19-related hospitalizations.

The manuscript is well written and straightforward. A few minor comments are provided below.

1) Introduction (page 3): the international readership might find some demographic context on the region of Ottawa useful, as well as on the size of the COVID19 outbreak in the region.

2) Methods section (page 3): please define REB.

3) Results section (pages 4-5): why would family medicine practitioners find themselves in the position of having to allocate a ventilator?

4) Discussion: The Authors state in the Introduction that a regional guideline on triaging acces to critical care beds was issued, which is referenced. A brief comment on the guideline and the criteria selected for resource allocation might enrich the discussion, especially in light of the survey's findings.

6. PLOS authors have the option to publish the peer review history of their article (what does this mean?). If published, this will include your full peer review and any attached files.

Reviewer #1: No

Reviewer #2: No

Reviewer #3: **Yes: **Haitham Jahrami

Reviewer #4: No

Reviewer #5: **Yes: **Costanza Vicentini

---

## [Author Response · Author response to Decision Letter 0]

14 Aug 2020

Dear Editors,

Thank you for taking the time to review this paper. Below, please find our responses to the comments made by the peer reviewers. 

Editorial comments: 

1. The methods must be described in sufficient detail for other researchers to reproduce the methodology described. This is a major concern that needs to be addressed while submitting the revised draft of the manuscript. For further details kindly check the following website:

Response: Thank you for this comment. We have significantly added to the methods section and hope that these additions will allow other researchers to reproduce this methodology. Additionally, please see appendix 2 for an even deeper discussion of the methodology used to conduct this survey. 

Peer Reviewer Comments: 

1. I would recommend that the authors reformulate their conclusions such as to clearly demonstrate that they are evaluating the effects of resource insufficiency on physicians in Canada's healthcare sector, rather than evaluating whether physicians consider resource insufficiency to be present prior to the onset of the pandemic.

Response: Thank you for this comment. Our goal was indeed to examine the readiness for a projected situation of resource scarcity rather than to determine the respondent’s feelings on a current and demonstrable insufficiency of resources. We have amended our conclusion to bring this point forward and prevent potential misunderstanding. We have written: “This rapidly implemented survey provides insights into physician preparedness to make resource allocation decisions utilizing a series of hypothetical conditions related to COVID-19.”

2. Further, the researchers have relied on insufficient scholarship to demonstrate the veracity of their premise that resources are insufficiently available to provide care during COVID-19. The majority of the resources cited to this effect are newspaper articles and ethics or decision-science articles that assess how a lack of resources should be addressed. If available, more scholarship demonstrating the lack of healthcare sector resources in the face of Canada's COVID-19 pandemic should be included.

Response: Thank you for this comment. Our desire was not to demonstrate that there was already or would be a situation of resource scarcity, but to explore how physicians in our health system might respond to similar conditions experienced elsewhere in the world. A number government initiatives addressed this hypothetical circumstance, not least of which were the provincial guidelines discussed in the conclusion. Since that time, a more extensive literature has developed and thus we have included Barrett (2020) and Shoukat (2020) to the introduction (references 6 and 7) to provide a more scholarly background to the projections of resource scarcity in Ontario and Canada at the time. 

3. Second, the methodology section (p. 3) is very sparse, and while the methodology selected is clearly described in the ensuing body paragraphs, a more expansive description of the selection criteria, study design, and the perceived strengths and weaknesses of the methods employed would strengthen the article and enable the reader to critically assess how well the study results can be generalized.

Response: Thank you for bringing this to our attention. We have rewritten the methods section to include much more in-depth discussion of the survey methodology alongside a discussion in the conclusion section of the strengths and weaknesses of this methodology. We hope this will maximize the ability for readers to generalize our results. 

4. Further, I consider the following conclusion to be overstated or impossible to draw from the data collected:

"Respondents demonstrated a strong appetite for transparently-developed, evidence-based, and clear-to-follow guidelines to inform resource allocation decisions."

I assume that the above conclusions were drawn from the responses to the questions on pages 5, 6, and 7 of the survey questionnaire. I consider it to be appropriate to draw conclusions therefrom as to the desire of physicians to rely on policy guidelines as opposed to personal, value-driven approaches in making decisions regarding the allocation of limited resources. However, the questions asked in the survey, in my view, do not permit the drawing of conclusions regarding the content that respondents hope to see reflected in such resource-allocation policies.

Response: Thank you very much for this response. We certainly do not want to overstate our conclusions. This conclusion was drawn from survey question 6A, which asked “If you were writing a guideline on resource allocation in the time of a pandemic, what would you say is the most important thing to include?”. We have now included a table (Table 2) of the most common responses in the policy recommendations section to clarify how we came to these conclusions. 

5. I note that there is a significant over-representation of laboratory medicine specialists among respondents. It is not required, but add to the depth of insight provided by the study, for respondents to note if any correlations appear among their results regarding the specializations studied. More generally, considerable demographic information appears to have been collected about the participants, but no conclusions are drawn with regard thereto. Resources and space permitting, the authors should consider adding information with regards to the correlation emergent from this data.

Response: Thank you for this comment. We apologize if there was confusion but we report no respondents from laboratory medicine. Analyses for correlation between specialty and responses were performed, but did not reach the level of statistical significance. We have noted this in the results section. 

6. Last, the results provided appear to suggest that research participants would slightly favor the allocation of resources to Canadian citizens over non-Canadian citizens. This could have serious implications for the potential for Canada's pandemic response and potential systemic discrimination or bias - the researchers should consider highlighting this conclusion and its implications for pandemic response in their findings.

Response: Thank you for this suggestion. We have rewritten our discussion of this point: A consistent set of factors emerged as being important to most physicians' decision-making around resource allocation: survival likelihood, cognitive function, comorbidities and daily function. However, significant disagreement existed around other factors including age and citizenship, and our ability to draw conclusions about these factors is limited. In the absence of agreement and standard practices, physicians are at risk of being influenced by unconscious biases in the way resources are rationed. This tendency may be exacerbated when information about the patient or the clinical scenario is limited. 

7. The authors note that “At the time the survey was launched, there were approximately a dozen patients with COVID-19 admitted to The Ottawa Hospital,” which preceded the anticipated surge. Compared to overwhelmed hospitals in, for example, New York City, Lombardy, Madrid, etc. the survey was conducted at a time where the hospital (which is the largest tertiary center in the region) was far from being overwhelmed, as was intended. Can the authors comment on whether surveys will be conducted later in the outbreak or after the pandemic to assess changes in average perceptions, whether as a result of lived experiences or introduced policy changes, support mechanisms, and/or crash courses supporting these decisions?

Response: Thank you for this excellent suggestion! Unfortunately, this project was funded by a one-off grant, and as such funding is currently unavailable for follow-up projects. However, should funding become available, follow-up work, especially targeting physicians in populations where there was a demonstrated scarcity of resources, would be a valuable and interesting.

8. Interestingly, surgical respondents were found to be among the most ill-prepared in terms of making difficult resource allocation decisions. The authors may want to discuss this point, as 1) many countries include brief critical care training in surgical training and 2) surgical teams are commonly deployed as one of the first services to support COVID-19 care (e.g., because of reduced elective services, to place cathethers or perform tracheostomies, etc.), which may be currently or in the future perceived in Canada as well in case of major surges/outbreaks.

Response: Thank you for this interesting suggestion. Ultimately the difference between the surgeons and the other groups were not statistically significant, and therefore we do not believe it is warranted to draw conclusions from this finding. We have made this point clearer by adding the line: No other statistically significant correlations were found with regards to specialty. 

9. 1. Abstract background and body introduction: write abbreviations (here: SARS and COVID-19) in full when using them for the first time.

2. Methods: “REB”, “HSD”, and “CHERRIES” may be unclear to the reader, write abbreviations in full before using them.

3. Results, line 2: “to prevent data duplication” should be “to minimize data duplication”; including surveys that are 50% complete may still result in duplication

Response: Thank you for this suggestion. We have amended our manuscript to ensure that abbreviations are defined on their first use, and have clarified this line in the results section as our original wording may have been misleading. 

10. The charts can be visually improved using professional software!

Response: Thank you for your comment. Figures and tables were produced according to journal specifications and will be made available in the highest possible resolution. 

11. The manuscript addresses some challenging issues that physicians encounter while managing COVID-19 patients with limited resources. The questionnaire seems adequate. The language overall was adequate. I do recommend the study included only those who completed 66 to 75% of questionnaires.

Response: Thank you for this suggestion that we exclude respondents who completed less than 75% of the questionnaire. However, we respectfully disagree that this will meaningfully add to our analysis as it will only serve to reduce the number of responses available and ultimately remains an arbitrary threshold. 

12. Introduction (page 3): the international readership might find some demographic context on the region of Ottawa useful, as well as on the size of the COVID19 outbreak in the region.

Response: Thank you for this comment. We have included background demographic context on the Ottawa region as well as information on the expected COVID-19 outbreak in the introduction section. In early April 2020, we launched a survey of physicians practicing in the region of Ottawa, Ontario, Canada. Ottawa is the national capital, a cosmopolitan and bilingual city of approximately 1 million.

13. Methods section (page 3): please define REB.

Response: Thank you for this comment. We have defined REB and other acronyms when they are first introduced. 

14. Results section (pages 4-5): why would family medicine practitioners find themselves in the position of having to allocate a ventilator?

Response: Thank you for your question. In our region, physicians trained in family medicine often work in the emergency department and in palliative care, both areas where they would be directly involved in COVID-19 related care.

15. Discussion: The Authors state in the Introduction that a regional guideline on triaging access to critical care beds was issued, which is referenced. A brief comment on the guideline and the criteria selected for resource allocation might enrich the discussion, especially in light of the survey's findings.

Response: Thank you for this comment. We have included background information on the guidelines in the introduction. Under hypothetical “surge” conditions that were ultimately never met, patients would be excluded from critical care according to disease-specific mortality thresholds that become increasingly more stringent as resources become more limited.

---

## [Editor Report · Decision Letter 1]

26 Aug 2020

Physician Preparedness for Resource Allocation Decisions Under Pandemic Conditions: A Cross-Sectional Survey of Canadian Physicians, April 2020

PONE-D-20-15128R1

Dear Dr. Shamy,

We’re pleased to inform you that your manuscript has been judged scientifically suitable for publication and will be formally accepted for publication once it meets all outstanding technical requirements.

Kind regards,

Ritesh G. Menezes, M.B.B.S., M.D., Diplomate N.B.

Academic Editor

PLOS ONE

---

## [Editor Report · Acceptance letter]

14 Oct 2020

PONE-D-20-15128R1 

Physician Preparedness for Resource Allocation Decisions Under Pandemic Conditions:
A Cross-Sectional Survey of Canadian Physicians, April 2020 

Dear Dr. Shamy:

I'm pleased to inform you that your manuscript has been deemed suitable for publication in PLOS ONE. Congratulations! Your manuscript is now with our production department. 

Kind regards, 

on behalf of

Prof. Dr. Ritesh G. Menezes 

Academic Editor

PLOS ONE